# MASLD Under the Microscope: How microRNAs and Microbiota Shape Hepatic Metabolic Disease Progression

**DOI:** 10.3390/ijms26178633

**Published:** 2025-09-04

**Authors:** Clelia Asero, Maria Stella Franzè, Irene Cacciola, Sebastiano Gangemi

**Affiliations:** 1Division of Medicine and Hepatology, University Hospital of Messina, 98124 Messina, Italy; clelia.asero@gmail.com (C.A.);; 2Department of Clinical and Experimental Medicine, University of Messina, 98124 Messina, Italy; 3Division of Allergy and Clinical Immunology, University Hospital of Messina, 98124 Messina, Italy

**Keywords:** metabolic dysfunction-associated steatotic liver disease, microbiota, microRNAs, exosomes, artificial intelligence

## Abstract

Metabolic dysfunction-associated steatotic liver disease (MASLD) is currently the most prevalent cause of chronic liver disease worldwide. Its pathogenesis is complex and not yet fully elucidated but is commonly explained by the “multiple hit” hypothesis, which suggests that pathological behaviours interact with an unfavourable genetic background and the presence of cardiovascular comorbidities. Recent evidence has highlighted a potential role of the gut microbiota in the onset and progression of MASLD to metabolic dysfunction-associated steatohepatitis (MASH) and hepatocellular carcinoma (HCC), potentially driven by epigenetic modifications mediated by microRNAs (miRNAs). MiRNAs are small, non-coding RNAs that regulate gene expression both intra- and extracellularly. Notably, emerging data suggests a bidirectional communication between the gut microbiota and the host, mediated by miRNAs via exosomes and outer membrane vesicles. The primary aim of this review is to explore the epigenetic crosstalk between the host and the gut microbiota through miRNA expression, with the goal of identifying specific pathways involved in MASLD development and natural history. A secondary objective is to evaluate the potential applications of artificial intelligence in the analysis of these complex host–microbiota interactions, to standardize the evaluation of microbiota and to create a model of the epigenetic changes in metabolic liver disease.

## 1. Introduction

Metabolic dysfunction-associated steatotic liver disease (MASLD) nowadays represents the most common cause of chronic liver disease worldwide; it affects over 38% of general adult population but its prevalence is expected to increase up to 55.4% by 2040 [1,2,3]. According to current international guidelines, MASLD is characterized by an excessive triglycerides’ storage in the liver, associated with at least one cardiometabolic risk factor, such as pre-diabetes or type 2 diabetes (T2D), overweight or obesity, dyslipidemia, and arterial hypertension, in the absence of other manifest causes of liver disease [4,5,6,7].

MASLD pathogenesis is a complex process and not fully understood [8]. Literature data propose the “multiple hits” hypothesis, suggesting that MASLD could be the consequence of a profound systemic disturbance in lipidic and glycemic metabolism [9,10]. In this context, insulin resistance (IR) seems to play a central role, because it enhances liver de novo lipogenesis, but also determines a dysregulation in lipidic metabolism in the visceral adipose tissue, leading to an imbalance in liver lipid storage and clearance [11]. Furthermore, this chronic metabolic disturbance triggers inflammation cells and determines cytokine and reacting oxygen species (ROS) release, a process known as lipotoxicity, which worsens IR itself and leads to liver fibrosis via the activation of the hepatic stellate cells into myofibroblast. In this process, genetic factors and lifestyle behaviors both play a crucial role [12]. In fact, while polymorphisms like patatin-like phospholipase domain containing protein 3 (*PNPLA3*), glucokinase regulator gene (*GCKR*), membrane-bound O-acyltransferase domain containing 7 gene (*MBOAT7*), farnesyl-diphosphate farnesyl transferase 1 gene (*FDFT1*), and transmembrane 6 superfamily protein 2 gene (*TM6SF2*) variants are responsible for the genetic background of the disease, promoting triglycerides accumulation and enhancing de novo lipogenesis, high-fat diets, excessive fructose intake, physical inactivity, and alcohol consumption also contribute to its development and progression [13,14,15,16] (Figure 1). Fructose intake, in particular, represents a major risk factor for MASLD onset, as it contributes to metabolic imbalance through multiple hormonal disturbances, including increased appetite and reduced insulin secretion. Moreover, it promotes inflammation via ROS generation, hyperuricemia, hypertriglyceridemia, arterial hypertension, and contributes to gut dysbiosis [17].

MASLD can progressively evolve into metabolic dysfunction-associated steatohepatitis (MASH), characterized by hepatocellular ballooning and lobular inflammation, but also into liver cirrhosis and hepatocellular carcinoma (HCC) [18,19]. Interestingly, the general outcome of MASLD patients is not only influenced by the status of liver disease but also by the presence of cardiovascular risk factors, which increase the risk of disability and mortality for major cardiovascular events [11,20]. In fact, some authors consider MASLD as the hepatic expression of metabolic syndrome, which is defined by the presence of three associated metabolic risk factors such as hypertriglyceridemia (triglycerides ≥ 1.7 mmol/L), low HDL-cholesterol levels (<1 mmol/L in males and <1.3 mmol/L in females), central obesity (waist circumference > 102 cm in males and >88 cm in females), presence of hypertension (blood pressure ≥ 135/85 mmHg or medication), or fasting plasma glucose ≥ 6.1 mmol/L [21].

Recently, a great attention has focused on the role of the microbiota in the development and progression of MASLD, as several studies have highlighted its immunomodulatory effects [22,23]. Moreover, there is growing interest in whether microbiota can be epigenetically modulated through host-derived microRNA-mediated crosstalk, and if this interaction might influence the natural history of the disease or serve as a potential therapeutic target.

The aim of this review is to explore the role of microbiota in the pathophysiology of MASLD, assess whether it can be modulated by microRNAs, and discuss the potential therapeutic implications of this interaction. A secondary aim is to explore how artificial intelligence could be applied to the study of these complex host–microbiota interactions, and to the creation of a precise model of the epigenetic pathways.

For this purpose, systematic research on PubMed has been performed, using the following terms as guided by MeSH database: “Non-alcoholic Fatty Liver Disease”, “microbiota”, and “microRNAs”. Scientific articles published over a time span of approximately 10 years, with a single exception, have been selected. International reviews and guidelines were mainly used in drafting the introductory sections, whereas clinical studies of observational or experimental nature, both in vitro and in vivo, were employed to support the hypotheses concerning the interactions between microRNAs (miRNAs) and microbiota.

## 2. microRNAs and Microbiota

MiRNAs are small non-coding RNAs, constituted by single-stranded RNAs with 19–23 nucleotides, which do not encode proteins but regulate gene expression and maintain genetic pathways stability, including those responsible for metabolic homeostasis [20,24,25,26]. In mammals, miRNAs bind with partial complementarity to the “seed region” within a binding site located in the 3′ untranslated region (3′ UTR) of target mRNAs. However, this interaction is not specific to a single target; in fact, a single miRNA can regulate multiple targets, thereby influencing various gene pathways. Moreover, miRNA expression levels have been shown to vary under different pathological conditions and, even within the same disease, can differ depending on the stage of disease progression [27]. These findings suggest that miRNAs may play a role in the pathogenesis and clinical course of various diseases.

MiRNAs act as transcriptional or post-transcriptional genes regulator within the nucleus, but they also circulate in biological fluids, such as blood, transported by exosomes [13,28]. Exosomes are small membrane-bound extracellular vesicles, ranging from 40 to 100 nm in diameter, present in almost all biological fluids and containing various nucleic acids, including miRNAs and other non-coding RNAs [29]. Current literature data suggests that exosomes and miRNAs could play a role in the interaction between hosts’ cells and gut microbiota, allowing a bidirectional communication that maintains homeostasis.

With the term “microbiota”, we refer to all the species of bacteria, virus, archaea, fungi and protozoa that colonize the different tracts of the digestive system. It is estimated that 90% of the cells in the human body are constituted by microbiota, which is nowadays considered the most complex ecosystem in nature [30]. More than 800 species of bacteria have been identified in a healthy bowel, with continue dynamic changes in bacteria quantity and composition in response to external factors, such as diet habits, water sources, physical activity, and diseases development [31,32]. Microbiota composition is determined by several physiological and pathological factors, including maternal diet during pregnancy, the modality of delivery, and breastfeeding. Moreover, environmental influences during adulthood play a significant role, causing microbiota changes throughout one’s lifetime [33,34]. Also, different microbiota species are present at various sites of the gastrointestinal tract. For example, *Firmicutes* and *Bacteroidetes* are the dominant bacterial phyla in the large intestine, and the *Firmicutes:Bacteroidetes* ratio was related to a susceptibility in several disease, despite the heterogeneity between individuals [35]. In healthy subjects, a mutualistic relationship exists between the host and his microbiota, a balanced state known as eubiosis. If this equilibrium is disrupted due to an alteration in the quantity or quality of the microbiota composition, dysbiosis occurs.

Microbiota can physiologically interact with the host, with various mechanisms that involve microRNAs [36]. In fact, host-derived microRNAs, transported via exosomes, have been shown to exert regulatory effects on the gut microbiota. Conversely, bacterial microRNAs are enclosed within the outer membrane vesicles (OMVs), enabling them to interact with the host, regulating host gene expression and translation [37]. This dynamic interplay is relevant due to its immunomodulatory properties and could be implicated in the pathogenesis of numerous diseases but also influence therapeutic responses, giving exciting possibilities for understanding disease mechanisms and potential therapeutic applications [23,38,39].

## 3. How Are microRNAs Involved in MASLD Onset?

In MASLD, several miRNAs have been identified to explain the disease natural history, and, in particular, the development of MASH and HCC [40,41]. It has been noticed that miRNAs expression greatly varies during the diseases’ natural course. Particularly, some miRNAs are overexpressed—such as *miR-379*, *miR-451*, *miR-129*—while others like *miR-29a* are downregulated; others appear to be significantly related to MASH development—such as *miR-144*, *miR-33a*, *miR-122*, *miR-24a*, *miR-21* and *miR-132*—or HCC onset (*mi-22*, *miR-6a*, *miR-26a-1*, *miR-192*, *miR-122* and *miR-125b*) [42].

Among them, *miR-34a* is overexpressed in both MASLD and MASH, but also in T2D, compared to healthy controls. The enhanced activity of *miR-34a* depends on the regulation of Farnesoid X receptor (FXR), Small Heterodimer Partner (SHP), and p53 activity [13]. It has been hypothesized that *miR-34a* balances energy homeostasis and lipidic metabolism through sirtuin 1 (SIRT1) expression. Specifically, *miR-34a* activity influences lipidic absorption, but also inflammation, fatty acids oxidation and apoptosis. In the hepatocytes, *miR-34a* inhibit SIRT1, contributing to steatosis development via Sterol regulatory element binding proteins (SREBPs), Peroxisome proliferator-activated receptor-gamma coactivator-1 alpha (PGC-1α), Liver X Receptors (LRXs) and forkhead box protein O1 (FOXO-1) [13,43,44]. Additionally, *miR-34a* regulates 3-Hydroxy-3-Methyl-Glutaryl-Coenzyme A (HMG-CoA) activity, and its overexpression increases endogenous cholesterol synthesis, contributing to steatosis development. *MiR-34a* overexpression in MASLD impairs hepatic lipophagy, facilitating lipid droplets accumulation, and activates SIRT1/p53 pro-apoptotic pathway via the enhanced expression of p66Shc, which converts oxidative stress signals into apoptosis. These hypotheses are confirmed by the experiments on *miR-34* antagonism, in which there is a normalization of SIRT1 activity, a downregulation of de novo lipogenesis, and an increased βoxidation in the liver.

Another key miRNA related to MASLD development is *miR-122*, the first identified as a regulator of metabolic homeostasis. Primarily expressed in the liver, it is the major regulator of lipidic metabolism, and it is also responsible for hepatocyte differentiation [45]. Its role in MASLD development is debated, due to conflicting clinical data. Some authors suggest that *miR-122* levels are increased in MASLD subjects compared to healthy controls [46], but these levels gradually decrease from MASLD to MASH to liver fibrosis [47]. In *miR-122* knockout mice, the absence of *miR-122* was related to the development of hepatic steatosis, steatohepatitis, and HCC. Also, *miR-122*^−/−^ mice exhibit increased cholesterol synthesis alongside decreased very-low-density lipoprotein (VLDL) secretion. However, in vitro studies on *miR-122* inhibition and overexpression presented conflicting results, leading to both a pro- but also an anti-steatotic role of the microRNA in the hepatocytes [13]. Some research found a positive correlation between serum *miR-122* and Interleukin 1 alpha (IL-1α), but also with lobular inflammation and hepatocellular ballooning. In these studies, *miR-122* silencing reduced endogenous cholesterol synthesis, and increases β-oxidation in the liver, with a consequent reduction in cholesterol, triglycerides and steatosis levels [43]. Conversely, other studies suggest the opposite, affirming that *miR-122* downregulation was related to MASH and fibrosis via the HSC/NEAT1 pathway, and the activation of Krüppel-Like Factor 6 (KLF6), which is a pro-fibrotic transcriptional factor. Also, literature data confirm that *miR-122* knockout mice exhibit a high production of pro-inflammatory cytokines and chemokines, but also an increase in Nuclear Factor κB (NF-κB) activity, which could be responsible for MASH development. Thus, further investigations are needed to explore the exact role of this miRNA in MASLD natural history.

Among the upregulated miRNAs in MASLD, there is *miR-33*, which is one of the microRNAs responsible for cholesterol and fatty acid metabolism. It targets the cholesterol efflux regulatory proteins ATP-binding cassette transporter A1 (ABCA1) and ATP-binding cassette subfamily G member 1 (ABCG1) and regulates β-oxidation via Carnitine Palmitoyltransferase 1A (CPT1A) and adenosine monophosphate-activated protein kinase α (AMPKα) modulation. Literature data suggest that this microRNA is also connected with the development of atherosclerosis, cardiovascular diseases, and obesity, which contribute to worsen the general outcome of MASLD patients [40].

Studies on hepatic miRNAs in MASLD, applying microarrays technique on liver biopsies, identified specific miRNAs for MASH, with some of them upregulated (*miR-224*, *miR-34a*, *miR-200a*, *miR-146*, *miR-222*), while others are downregulated (*miR-617*, *miR-641*, *miR-198*, *miR-765* and *miR-601*). While *miR-26a* appears downregulated in MASLD due to the chronic inflammatory stress, *miR-378*—which promotes hepatic inflammation via the NF-κB/Tumor Necrosis Factor alpha (TNF-α) signaling—and *miR-144*—which impairs antioxidant response to lipid accumulation via nuclear factor erythroid 2-related factor—are overexpressed in MASH [48,49,50]. A single study examinates the differences in miRNA expression across the entire spectrum of MASLD, from simple steatosis to F3–F4 fibrosis, despite the small number of patients. According to this research, *miR-375* expression was downregulated in cirrhosis vs. NASH, while three miRNAs’ levels gradually increase (*miR-301a-3p* and *miR-34a-5p*) or decrease (*miR-375*) in the progression from MASLD to cirrhosis [13].

Interestingly, some miRNAs present a specific activity both inside and outside the liver determining MASLD progression and energy metabolism impairment. In murine models, miRNAs seem to be involved in the communication between inflammatory cell and hepatocyte, but also between adipose tissue, skeletal muscle and hepatocyte, impairing lipid storage and metabolic balance beyond the liver. Among microRNAs mostly involved into metabolic dysregulation, *miR-122*, *miR-192* and *miR-375* must be mentioned. Specifically, *miR-192* promotes the activation of myofibroblast targeting Peroxisome proliferator-activated receptor gamma (PPARγ), activates the hepatic stellate cells in response to lipotoxicity, promotes the expression of cytokines and facilitates the activation of M1 macrophages, especially in MASH models.

## 4. How Can Microbiota Influence MASLD Pathogenesis?

Several mechanisms have been proposed to explain how microbiota could impact on MASLD progression, despite the results are still controversial [51]. Current literature data sustained that in both MASLD and Metabolic dysfunction and alcohol-related liver disease (MetALD), there is a bacterial overgrowth in the digestive tract, along with significant alterations in microbiota composition, which develop in the early stages of the disease [52]. These changes in microbiota influence gut permeability through a reduction in zona occludens 1 (ZO-1) protein, a key tight junction protein [53]. Simultaneously, inflammatory molecules such as peptidoglycan, lipopolysaccharides (LPS), ammonia, phenols, and acetaldehyde are released in large quantities, easily reaching bloodstream due to the increased gut permeability, while butyrate levels decrease [54]. These pro-inflammatory molecules reach the liver through the portal vein, where determine the activation of the resident immune cells—lymphocytes, macrophages, dendritic cells and natural killer cells—through the pathogen-associated molecular patterns (PAMPs) and damage-associated molecular patterns (DAMPs) pathway, with the consequent activation of the stellate cells and the induction of a local pro-inflammatory state. Moreover, LPS further promotes the release of pro-inflammatory cytokines such as TNF-α, Interleukin 1β (IL-1β) and Interleukin 6 (IL-6), as well as chemokines through Toll-Like Receptor 4 (*TLR4*) in endothelial cells and Toll-Like Receptor 9 (*TLR9*) in dendritic cells, enhancing liver damage and activating stellate cells [55] (Figure 2). Additionally, LPS contributes to increase fat consumption, with the consequent release of free fatty acids that could bring to MASLD development. The liver pro-inflammatory state and the large amount of ROS chronically released, could be involved in cellular DNA damage, potentially playing a role in HCC development. Also, gut microbiota affects hepatic lipid metabolism and de novo lipogenesis via the FXR and the Takeda G-protein coupled receptor 5 (TGR5) pathways, but it also exacerbates insulin resistance onset through butyrate-producing bacteria activity and choline deficiency [16]. Furthermore, gut microbiota regulates energy intake from dietary food through different mechanism, including bacterial enzyme extraction of calories from dietary polysaccharides and modulation of lipoprotein lipase activity to facilitate fat storage [56]. Finally, in MASLD, there is a reduction in the abundance of bacteria responsible for the conversion from primary bile acids to secondary bile acids, that further contribute to dysbiosis [57].

However, the relationship between microbiota and host is not so oversimplified and seems to involve bacterial and human genetic mechanisms. Studies on murine models, highlighted that inflammasome-mediated dysbiosis could drive the progression from MASLD to MASH [35]. Additionally, bacterial DNA could activate inflammatory signaling via *TLR9*, which led to the secretion of Interleukin 12 (IL-12) and TNF-α through the NF-κB/Mitogen-activated protein kinase (MAP-kinase) pathway. Otherwise, *TLR9* could activate the IκB kinase-α and contribute to type I interferon release, which is a possible molecular mechanism implied in MASH development [55,58,59]. Focusing on human microbiota, DNA genome sequencing from the stool of MASLD patients has revealed significant microbiota shifts as liver disease progresses from MASLD to MASH to liver cirrhosis. In fact, while *Firmicutes* and *Eubacterium rectale* were prevalent in MALSD, *Bacteroides*, *Allisonella* and *Parabacteroides* are increased in MASH, and *Proteobacteria*, *Ruminococcus* and *Escherichia coli* (*E. coli*) characterized fibrosis [60]. Several studies highlighted that the number of Gram-negative bacteria progressively increase as MASLD advances toward fibrosis [61]. Specifically, it has been hypothesized that *Proteobacteria* contribute to ethanol production, elevating oxidative stress, which is a key feature of MASH [62]. Other studies identify a prevalence of *Klebsiella* in MASLD and an abundance of *Escherichia*, *Bilophila* and *Rodobacter* in MASH patients. Interestingly, the differences in the results proposed are justified by the variability in the study analyzed, which differ greatly for patients, geographical areas and dietary habits. The variability in results across different studies, along with the ongoing fluctuations in an individual’s microbiota over time [63], are the main reasons why such changes are not currently used as reliable markers of MASLD [64].

## 5. miRNAs and Microbiota: Is There a Crosstalk in MASLD Natural History?

Research on microbiota in MASLD has led experts to question whether the use of probiotics could improve steatosis and liver inflammation, and promising data came from the use of *Lactobacillus Casei* (*L. casei*) [65,66]. In MASLD, it is not observed an absolute reduction in *Lactobacilli* quantity; however, it has been hypothesized that *Lactobacilli* activity may decline during the progression of the disease. Numerous studies on murine models have shown that supplementation with *L. casei* reduces steatosis degree and hepatic inflammation. Moreover, in some experimental models, *L. casei* supplementation has led to a significative improvement in MASH [67,68]. These findings highlight the potential role of probiotic-based interventions in treating MASLD and MASH, although further clinical trials are needed to confirm their effectiveness in human patients. In this context, miRNAs expression could play a decisive role. Notably, studies highlighted that supplementation in *L. casei* is related to a reduction in *miR-144* expression. This downregulation led to a significative amelioration in gut permeability, through an increase in the expression of ZO-1 and Occludin protein (OCLN) proteins [69]. It could be hypothesized that the improvement in the gut permeability, mediated by *miR-144* downregulation, contributes to reducing steatosis grade and liver inflammation. In fact, *miR-144* is involved in MASH development through a positive regulation of Toll-like receptor 2 (*TLR2*) gene expression: *miR-144* downregulation and the consequent reduction in *TLR2* during probiotics treatment is reflected in the reduction in cytokines levels—such as IL-1β, TNF-α, IL-6 and Interleukin 8 (IL-8)—highlighted in the literature [70]. Also, oral administration of *L. casei* has been shown to restore the *Firmicutes:Bacteroides* ratio, suggesting that *Lactobacilli* may improve liver disease in various ways [71]. Another interesting consideration derives from in vitro models, in which *L. casei* appears to regulate in cholesterol metabolism. Quantitative reverse transcription polymerase chain reaction (qRT-PCR) analysis showed that *L. casei* decreased hepatic fatty acids synthesis, reducing fatty liver accumulation, inhibiting the expression of Sterol regulatory element binding protein 1c (SREBP-1c) and Acetyl-CoA carboxylase (ACC), but also increasing Peroxisome proliferator-activated receptor alpha (PPARα) expression with the consequent fatty acid catabolism [72,73,74]. This datum is in line with our analysis; in fact, *miR-144* inhibit cholesterol export from the hepatocytes via the downregulation of ABCA1 and ABCG1, with a consequent lipidic accumulation. *MiR-144* silencing via *L. casei* could restore ABCA1/ABCG1 activity, ameliorating cholesterol metabolism into the hepatocyte [75] (Table 1). Overall, all these findings support the potential use of *L. casei* in daily clinical practice, alongside lifestyle-changing, lipid-lowering, and hypoglycaemic drugs, contributing to the amelioration of liver disease natural history. However, more data, particularly in human models with validated clinical trials, are needed to further confirm this hypothesis.

Among miRNAs more frequently associated with MASLD, there is *miR-21*, which is involved in each phase of the disease progression. In fact, it regulates lipid metabolism, enhancing lipid accumulation via sterol regulatory element binding protein 1 (SREBP1), 3-Hydroxy-3-Methylglutaryl-Coenzyme A Reductase (HMGCR), and Fatty Acid binding protein (FABP7), but it is also implicated in IR onset through FOXO-1, insulin induced gene 2 (*INSIG2*), Signal Transducer and Activator of Transcription 3 (STAT3) and Hepatocyte nuclear factor 4 alpha (HNF4-α). Also, *miR-21* induces inflammation and fibrosis development via the inhibition of PPARα, and it appears to be involved in carcinogenesis [76,77]. According to the literature data, *miR-21* is upregulated in MASH and liver cirrhosis, and it is expressed in both hepatocytes and non-parenchymal liver cells, including hepatic stellate cells, biliary and inflammatory cells [13,43]. It seems to be directly involved in inflammation and fibrosis development, because it induces hepatic stellate cells and myofibroblast activation via Extracellular signal regulated kinase signaling (ERK-signaling) but also stimulating Wingless/Integrated beta catenin signaling (WNT/b-catenin signaling). In silico analysis demonstrated that *miR-21* is responsible for the modulation of some microbiota bacteria, such as *Oscillobacter*, *Prevotella* and *Ruminococcus* [38]. Specifically, in MASH, there is a higher concentration of *Ruminococcus* and an enhanced activity in *miR-21*, which led to abnormalities in lysin degradation pathway and fatty acids elongation in mitochondria. Specifically, in mammals, lysine is metabolized by microbiota in Acetyl-Coenzyme A (CoA), and excessive CoA levels impair cellular autophagy, potentially contributing to MASH and fibrosis development (Table 1). Once again, restoring microbiota balance may play an epigenetic role in metabolic disease progression, offering new strategies for disease management, despite further investigation being needed.

## 6. Artificial Intelligence and Its Application to MASLD Study: Future Perspectives

Studies on microbiota are inherently complex due to the high variability within the sample population and the presence of numerous confounding factors, which interfere with standardization [78]. Similarly, the analysis of microRNA activity represents a considerable challenge, as a single miRNA can target multiple genes, and conversely, a single gene may be regulated by various miRNAs [24]. In this context, artificial intelligence (AI) could provide practical support for data analysis and error detection, enabling a more accurate evaluation of a large amount of information. Indeed, artificial intelligence applications in medicine primarily focused on outcome prediction, variable categorization, detection—namely, the identification of abnormal features within a population—and dimensionality reduction, which involves the elimination of low-informative variables and the selection of the most relevant ones for the purpose of the analysis [79]. AI has already revolutionized clinical research by facilitating personalized medicine via the application of machine learning and deep learning techniques to analyze large-scale datasets, commonly referred to as “big data” [80]. According to a survey promoted by the European association for the study of the liver, AI demonstrated great potential in hepatology, with application ranging from image interpretations—both in radiology and histology—to the development of predictive models for the evaluation of chronic disease natural history [81,82,83]. It may also be used for risk stratification and the identification of specific biomarkers in liver diseases. Recently, there has been growing interest in the application of AI in the field of omics [84]. AI could be employed not only in miRNAs sequencing, but also in the identification of the related gene pathways and, consequently, in elucidating the biological effects of the miRNAs under investigation. This approach could lead to the development of specific and reproducible models which may allow the creation of gene-targeted therapies based on the silencing of specific miRNAs, with the aim of preventing disease onset or modifying its progression [85,86]. Similar considerations may be applied to the study of gut microbiota. The analysis of large amount of clinical data, associated with the stratification of our population based on the confounding variables, may allow the standardization of microbiota characteristics and to clarify its role in diseases pathogenesis [87,88]. Additionally, the characterization of disease-specific pathways could justify the use of targeted microbiota-modulating therapies—such as prebiotics, probiotics, and symbiotic, as well as fecal microbiota transplantation—that currently have a limited application in daily clinical practice [69,89]. Once again, this approach would facilitate personalized treatment strategies and contribute to the advancement of precision medicine, but further validation studies are necessary.

Despite the promising results achieved with AI applications in medicine, ethics concerns remain regarding the implementation of AI systems in clinical practice. In fact, AI presents inherent limitations and is not immune to bias, which can be either systematic or random [90]. In some instances, algorithms have utilised inappropriate or non-specific variables to generate output, leading to the development of flawed models. One of the major challenges in the clinical adoption of AI is the lack of transparency in its algorithms, particularly when black boxes models are employed. The inability to fully understand the AI decision-making process undermines trust in the outputs generated, thereby limiting the integration of these models into clinical practice. To address this issue, the concept of explainability has been introduced—defined as the ability to rationally and causally justify the analytical process employed by AI [91]. However, current explainability often provides incomplete or suboptimal insights, which is why in medical context, the use of transparent and interpretable algorithms is generally preferred to ensure clinical accountability and reliability.

In conclusion, artificial intelligence holds significant potential as a tool for analyzing large datasets, including those related to microRNAs and microbiota. However, its application must be approached with caution to minimize errors and address ethical concerns effectively.

## 7. Conclusions 

MASLD is a complex disease characterized by the simultaneous disruption of multiple metabolic pathways. While it is well established that lifestyle behaviors and cardiometabolic comorbidities play a fundamental role in the development and progression of the disease, recent studies on gut microbiota have provided fascinating insights into its pathogenesis and natural history. A deeper comprehension of microbiotas’ role in MASLD, as well as the associated epigenetic modifications mediated by miRNAs, could allow the development of novel targeted therapies, expanding the range of therapeutic options available to patients in routine clinical practice. In this context, artificial intelligence could provide valuable tools to standardized data analysis and create real-life models to deeply comprehend the disease’s natural history if wisely used.

## Figures and Tables

**Figure 1 ijms-26-08633-f001:**
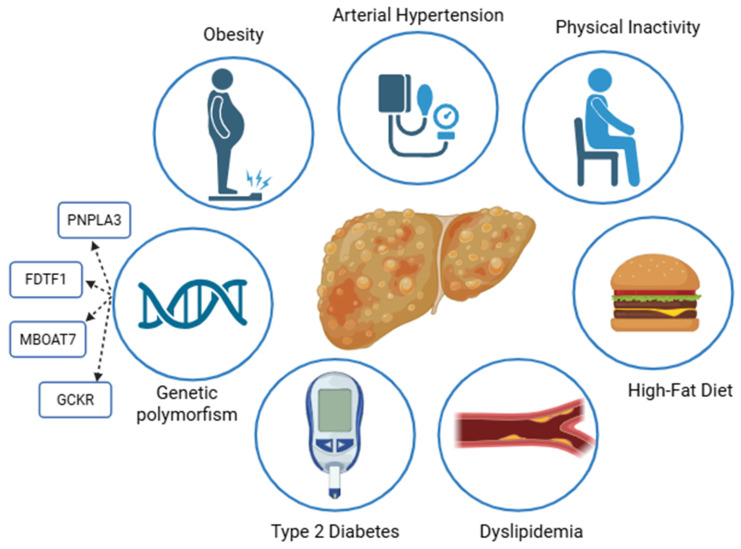
In MASLD, there is an excessive accumulation of triglycerides in the hepatocytes, associated with at least one cardiometabolic risk factor such as type 2 diabetes, obesity, arterial hypertension, or dyslipidemia [4,5,6,7]. Additionally, genetic polymorphisms—such as variants in *PNPLA3*, *GCKR*, *MBOAT7*, *FDFT1*, and *TM6SF2*—may contribute to its pathogenesis through an imbalanced lipid storage and synthesis, along with unhealthy lifestyle habits like physical inactivity and a high-fat diet [13,14,15,16]. Created in https://BioRender.com.

**Figure 2 ijms-26-08633-f002:**
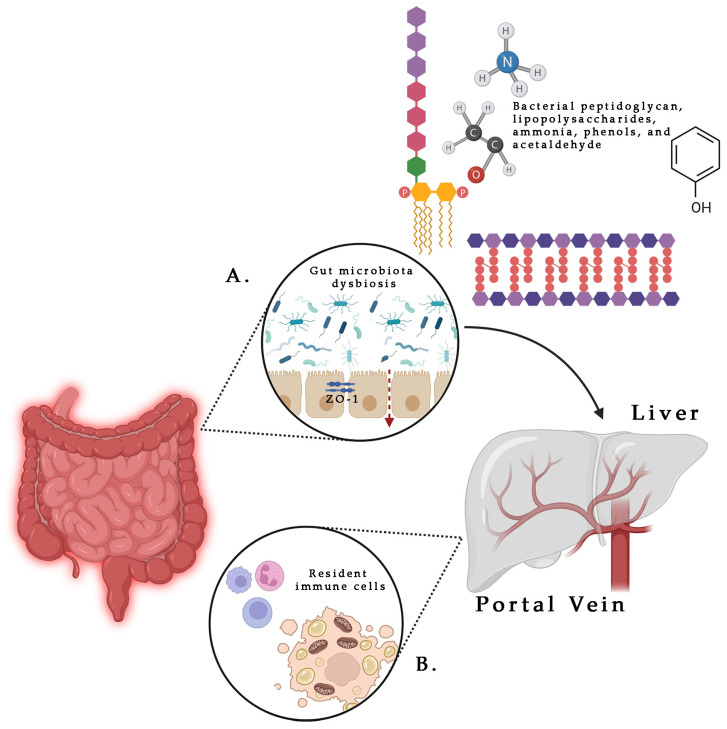
Interactions between gut microbiota (**A**) and liver immune system (**B**) in the physiopathology of MASLD. Gut microbiota abnormalities led to a reduction in ZO-1 protein that determines an increase in gut permeability (as indicated by the dotted arrow). Bacterial inflammatory molecules (peptidoglycan, lipopolysaccharides, ammonia, phenols, and acetaldehyde), released in large quantities into the bloodstream, reach the liver through the Portal Vein (as indicated by the continuous arrow) with the consequent activation of the resident immune cells and the induction of a local pro-inflammatory state. Created in https://BioRender.com.

**Table 1 ijms-26-08633-t001:** Interactions between microRNAs and microbiota in MASLD development and natural history. Legend: ↓ decreased expression; ↑ increased expression; (?) not specified if increased or decreased expression by available literature data.

Microbiota	microRNAs	Signaling Pathways	Biological Effects	References
↓ *L. casei*	↑ *miR-144*	↑ *TLR2* ↑ SREBP-1c ↑ ACC ↓ PPARα ↓ ABCA1 ↓ ABCG1	↑ IL-1β, TNF-α, IL-6, IL-8 ↓ ZO-1 ↓ OCLN Abnormalities in *Firmicutes:Bacteroides* ratio ↓ Fatty acids catabolism	[69,70,71,72]
(?) *Oscillobacter* (?) *Prevotella* ↑ *Ruminococcus*	↑ *miR-21*	↑ SREBP1 ↑ HMGCR ↑ FABP7 ↑ FOXO-1 ↑ *INSIG2* ↑ STAT3 ↑ HNF4-α ↓ PPARα ↑ ERK- signaling ↑ WNT/b-catenin ↑ CoA	↑ Lipid accumulation ↑ IR ↑ Inflammation Activation of stellate cells and myofibroblast Abnormalities in lysine degradation Abnormalities in fatty acids elongation Impaired autophagy	[38]

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
