# Peer review of "MASLD Under the Microscope: How microRNAs and Microbiota Shape Hepatic Metabolic Disease Progression"

_ijms, 2025, doi:10.3390/ijms26178633_

Round 1
Reviewer 1 Report
Comments and Suggestions for Authors
This review manuscript provides a timely and informative overview of the epigenetic interplay between gut microbiota and the host in the context of Metabolic Dysfunction-Associated Steatotic Liver Disease (MASLD). The authors focus specifically on the role of microRNAs (miRNAs) in mediating this crosstalk and propose the potential application of artificial intelligence (AI) to model these complex interactions. The topic is highly relevant, and the manuscript is generally well-structured and clearly written.
However, there are a number of issues—both conceptual and technical—that require correction before the manuscript can be considered suitable for publication. I outline my specific comments below. I recommend acceptance after moderate revision.
1.Language and Grammar: Several grammatical and typographical issues need attention throughout the manuscript. For instance, the phrase "Tool-Like Receptor" should be corrected to "Toll-Like Receptor" consistently, including in lines 220, 282, 326, and in the Abbreviations section.
2.Missing References: The statement “MiR-144 silencing via L. casei could restore ABCA1/ABCG1 activity, ameliorating cholesterol metabolism into the hepatocyte” lacks a proper citation. Please add a supporting reference or rephrase the claim accordingly.
3.Abbreviation Consistency: Some abbreviations are inconsistently formatted. For example:“ZO1” should be unified as “ZO-1” throughout the manuscript.“TLR2” vs. “TLR-2” should be standardized (preferably use the commonly accepted “TLR2” format). Please ensure all abbreviations are consistent across the text and figures.
4.Figure Annotation: Figures 1 and 2 currently contain only graphical elements with no embedded labels or textual annotations. While legends are provided, adding key labels directly within the figures would significantly improve clarity and reader understanding.
5.Gene Knockout Notation: The notation miR-122 -/- should be consistently formatted without spacing: miR-122−/−.
6.Quotation Marks: Ensure the use of quotation marks is typographically consistent across the manuscript. For instance, unify usage to standard English double quotes (“ ”) instead of mixed forms like "big data”.
7.Gene/Protein Names and Definitions: Terms such as SREBP1 (Sterol Regulatory Element-Binding Protein 1) should be uniformly spelled and defined throughout the manuscript. Similar inconsistencies were noted with other gene/protein names—please perform a full-text check and revise accordingly.
8.Reference Formatting: Several references exhibit formatting errors. For example, line 530 contains a formatting issue. Please carefully review and correct all references according to the journal’s citation style.
Reviewer 2 Report
Comments and Suggestions for Authors
The review article is very interesting and very well outlined; it provides new knowledge
I suggest minor changes:
1) Place the paragraph "How are miRNAs involved in MASLD onset" before the paragraph on miRNAs and microbiota, as this improves reading comprehension.
2)In the same paragraph, I suggest adding hard human data related to mi-RNA, such as Ghazy's results, which found that mi-RNA-122 at a cut-off ≥ 6.71 exhibited 53% sensitivity and 72% specificity to distinguish NAFLD from healthy controls (Clin Ter. 2025 May_Jun;176(3):344-349. doi: 10.7417/CT.2025.5232.)
In summary, the manuscript requires minimal changes for publication.
Reviewer 3 Report
Comments and Suggestions for Authors
The manuscript is a narrative review of MASLD, the primary aim of which is to map the relevant epigenetic host–microbiota interactions and discuss the potential of AI in the studies on MASLD. The manuscript is well-written and up-to-date, raising the currently emerging concern of liver steatosis. However, there are a number of issues that need to be addressed.
Methods: More detailed search criteria should be provided. Please include the time range, language and document type, etc. Maybe a figure that represents the selection process could be implemented into manuscript
Lines 54–55: Why is excessive fructose intake and altered fructose metabolism important in the development of MASLD? What is the role of fructose biotransformation by gut microbiota? Please explain.
Line 61: MASLD is often described as the hepatic manifestation of metabolic syndrome. A brief explanation of what MetS is and how it is diagnosed should be included in the context of CVD related to MASLD.
Lines 220, 282 and 326, as well as the reference list, should read 'toll-like receptor'.
On lines 246–255, the Authors state that 'several studies have highlighted', but no reference confirms this statement.
Lines 270–273: Are there only two studies on L. casei in MASLD/MASH improvement?
Lines 302 and 330: it should be 'SREBP1'.
Table 1. Please place relevant references in a separate column next to key findings in the table.
Figure 1: The genetic mechanisms of MASLD development should be highlighted in the figure caption.
Figure 2 is unclear. I cannot see the ZO-1 protein or the pro-inflammatory molecules entering the hepatocytes. The hepatocytes and the immune cells are not labelled. Please create a figure that clearly indicates the cause-and-effect relationship.
Reviewer 4 Report
Comments and Suggestions for Authors
This manuscript addresses a relevant topic on the interaction between miRNAs and gut microbiota in the context of MASLD. The paper is well organized, clearly written, and presents an engaging narrative. Addressing some points will improve clarity, scientific rigor, and the impact of the work:
- The abstract described AI as a secondary objective (line 26), but the introduction does not give any rationale, methodology, or integration into the study aims (line 73). AI is only mentioned again at the end of the paper, without relevant discussion. If AI is an objective, the authors should provide a background, justification, and methodological details. However, if AI is not central to the study, consider removing it.
- In the first paragraph (lines 34–35), the manuscript is confused about the prevalence and incidence: “…affects over 38% of the general adult population, but its incidence is expected to increase up to 55.4% by 2040.” This should be revised to distinguish between current prevalence and projected prevalence, or between prevalence and incidence if incidence estimation is the objective.
- Line 86 describes miRNAs as “constituted by double-stranded RNAs.” This could confound readers with miRNA biogenesis. Please clarify that mature miRNAs are single-stranded guide molecules derived from an initial double-stranded precursor (the miRNA duplex), where the passenger strand is typically degraded.
- Throughout the manuscript, the authors used “Tool-Like Receptor”. It should be corrected as “Toll-Like Receptor” (e.g., TLR-2, TLR-4, TLR-9).
- Some references cited for mechanistic statements (e.g., microbiota–miRNA interactions) are broad reviews rather than primary sources. Where possible, please cite primary research articles that directly support the mechanistic claims.
- Figures 1 and 2 are visually clear, but in my opinion, these figures do not give much real information. However, the reader would benefit from a statement indicating if the pathways are derived from experimental evidence, hypothesized mechanisms, or a combination of both.
- Certain sections, particularly regarding probiotic interventions (e.g., Lactobacillus casei effects), could be described with caution, emphasizing the preliminary nature of the evidence and the need for validation in well-designed human trials.
Round 2
Reviewer 3 Report
Comments and Suggestions for Authors
The Authors have addressed all of my comments. Now, I reccomend the presented manuscript for publication in IJMS.
Reviewer 4 Report
Comments and Suggestions for Authors
The authors have successfully addressed the suggestions and corrected the main issues. This paper provides valuable insights for the scientific community. In particular, the authors present the role of gut microbiota associated with MASH clearly and systematically.